# Wet-Spun Chitosan–Sodium Caseinate Fibers for Biomedicine: From Spinning Process to Physical Properties

**DOI:** 10.3390/ijms25031768

**Published:** 2024-02-01

**Authors:** Hazel Peniche, Ivy Ann Razonado, Pierre Alcouffe, Guillaume Sudre, Carlos Peniche, Anayancy Osorio-Madrazo, Laurent David

**Affiliations:** 1Ingénierie des Matériaux Polymères (IMP), Universite Claude Bernard Lyon 1, INSA de Lyon, Universite J. Monnet, CNRS, UMR 5223, 69622 Villeurbanne CEDEX, France; hazel@biomat.uh.cu (H.P.); icrazonado@up.edu.ph (I.A.R.); pierre.alcouffe@univ-lyon1.fr (P.A.); guillaume.sudre@univ-lyon1.fr (G.S.); 2Biomaterials Center, University of Havana, Havana 10600, Cuba; 3Faculty of Chemistry, University of Havana, Havana 10600, Cuba; peniche@fq.uh.cu; 4Laboratory of Organic and Macromolecular Chemistry (IOMC), Jena Center for Soft Matter (JCSM), and Center for Energy and Environmental Chemistry Jena (CEEC), Friedrich Schiller University of Jena, 07743 Jena, Germany; 5Laboratory of Organ Printing, University of Bayreuth, 95447 Bayreuth, Germany

**Keywords:** chitosan, casein, fiber wet spinning, mechanical properties, biomedical textiles

## Abstract

We designed and characterized chitosan–caseinate fibers processed through wet spinning for biomedical applications such as drug delivery from knitted medical devices. Sodium caseinate was either incorporated directly into the chitosan dope or allowed to diffuse into the chitosan hydrogel from a coagulation bath containing sodium caseinate and sodium hydroxide (NaOH). The latter route, where caseinate was incorporated in the neutralization bath, produced fibers with better mechanical properties for textile applications than those formed by the chitosan–caseinate mixed collodion route. The latter processing method consists of enriching a pre-formed chitosan hydrogel with caseinate, preserving the structure of the semicrystalline hydrogel without drastically affecting interactions involved in the chitosan self-assembly. Thus, dried fibers, after coagulation in a NaOH/sodium caseinate aqueous bath, exhibited preserved ultimate mechanical properties. The crystallinity ratio of chitosan was not significantly impacted by the presence of caseinate. However, when caseinate was incorporated into the chitosan dope, chitosan–caseinate fibers exhibited lower ultimate mechanical properties, possibly due to a lower entanglement density in the amorphous phase of the chitosan matrix. A standpoint is to optimize the chitosan–caseinate composition ratio and processing route to find a good compromise between the preservation of fiber mechanical properties and appropriate fiber composition for potential application in drug release.

## 1. Introduction

Polymeric fibers play an important role in the fabrication of biocompatible and biodegradable constructs. They are desirable for tissue engineering devices due to their large surface area, interconnectivity, open pore structure, and controlled mechanical properties. In addition, biodegradable constructs are also highly sought after for biomolecule delivery systems. Polymer fibers can be formed using a variety of spinning processes [1]. Bio-textiles (structures composed of textile fibers based on natural and/or synthetic biomaterials) are very common in tissue engineering. The applications of bio-textiles cover a large spectrum; they have been used in heart valve swing rings, vascular grafts, hernia repair meshes, percutaneous access devices, wound dressings, and various drug delivery systems [2].

Chitosan (CTS) is a cationic polysaccharide in acidic aqueous media and can form complexes through electrostatic interactions with biomacromolecules that carry negative charges, including anionic polysaccharides, nucleic acids, and particularly, proteins. The strength of these interactions depends on several factors, including the degree of acetylation (DA) and the molecular weight of chitosan, the pH (impacting on the charge density of chitosan), the ionic strength of the medium, the isoelectric point (IP) of the protein, and the hydrophobic/hydrophilic balance of the protein, among other intrinsic and extrinsic parameters [3]. Chitosan–protein materials have been prepared to harness the functional properties of the protein [4,5,6], to release the protein [7,8,9], or to increase the drug loading efficiency, i.e., to induce a delayed release or the protection of other active compounds [10,11,12]. Chitosan–protein associations may serve as biocompatible matrices or nanoparticles with functional properties for various applications in drug delivery or vaccine and food protection [13,14]. In the context of wet spinning, chitosan, as a semicrystalline polymer, can provide the ability to form stretchable hydrogel extrudates [15,16,17,18,19] to obtain dry fibers with fair mechanical properties [20,21]. Nevertheless, increasing the mechanical properties of chitosan fibers is technologically still challenging, but of key relevance for obtaining knittable fibers [22,23]. Various proteins and polysaccharide nanofillers have been utilized to enhance the mechanical properties of chitosan hydrogels and fibers through the association of opposite polyelectrolyte charges [17,24,25,26]. Recent research has focused on chitosan electrospun nanofibers and nanofibrous mats reinforced with silk fibroin (SF), resulting in promising materials for biomedical and tissue engineering applications [27,28,29,30]. Similarly, nanofibers were obtained from silk fibroin, chitosan, and gelatin blends using cross-linking agents [31]. Collagen–chitosan nanofibers associated with poly(vinyl alcohol) (PVA) or thermoplastic polyurethane (TPU) were also obtained by electrospinning to develop tissue engineering scaffolds [32]. Additionally, the use of other proteins such as silk sericin, antarctic krill protein, and gelatin [33,34,35] has been explored for the development of chitosan-based fibers with improved mechanical properties for various applications.

In preliminary works, we were able to prepare chitosan–albumin fibers by wet spinning mixture solutions of chitosan and bovine serum albumin (BSA) and their neutralization in a 0.1M NaOH coagulation bath without the use of cross-linking agents or post-treatments [36]. To our knowledge, prior to our study, there were few experiments on solution spinning of chitosan in combination with proteins. Our inspiration for this study came from the process of how cuticles are produced by insects and crustaceans, chitin interacting with cuticular proteins. We prepared bioinspired fibers from a complex of chitosan with BSA (model protein) by spinning chitosan–protein solutions. These fibers exhibited suitable mechanical properties for their application in tissue engineering, for example, annulus fibrosus repair. Moreover, it has been highlighted that chitosan–protein mixes may self-crosslink during stretching and drying [4].

Casein is the major ingredient of milk proteins and mainly consists of four phosphoproteins, αS1, αS2-, β-, and κ-casein, with weight ratios of 4:1:4:1, respectively. Their molecular weights range between 19 and 25 kDa, and the average isoelectric point (IP) is between 4.6 and 4.8 [37]. Caseins are amphiphilic proteins with high thermal stability, and the distinct hydrophobic and hydrophilic domains result in the self-assembly of casein molecules into stable micellar structures in aqueous solutions [38]. The four phosphoproteins interact and are maintained together through hydrophobic interactions and also by the bridging of calcium phosphate nanoparticles that bind to phosphorylated serine residues of the casein side chains [39]. A large proportion of κ-casein is located close to the micelle surface, providing steric stabilization by protruding out of the micelle with its hydrophilic glycosylated portions [40]. The stability of casein micelles is low at alkaline pH above 9, where the micelles disrupt into sub-micellar nanoclusters. Sub-micelles consist of a shell of casein proteins surrounding a calcium phosphate core nanoparticle, but they can be re-aggregated into larger micelles at lower pH [41]. Sodium caseinate is usually obtained after precipitation of casein in acidic media and further washing with sodium hydroxide. The amphiphilicity and specific peptide structure of casein proteins provide excellent emulsifying and gelling properties and ions/small molecules binding ability [42]. Additionally, casein and sodium caseinate can interact with biomolecules to form complexes and conjugates with a synergistic combination of properties [43]. Hence, the interaction between charged polysaccharides with milk proteins has been continuously exploited. Meanwhile, most studies have focused on the effect of pH, ionic strength, and bio-macromolecular structure on chitosan–protein association and phase behavior of chitosan–protein complexes [44,45,46,47].

Chitosan, casein, or caseinate and their associations have been developed for drug delivery, resulting in improved bioactivity and bioavailability, enhanced stability and water dispersibility, and sustained-release properties [47,48,49,50,51,52,53,54]. Ding L et al. investigated the interaction between chitosan and casein and constructed a detailed phase diagram of chitosan–casein for complexation [55]. They concluded that the critical chitosan concentrations for stable complexes were dependent on pH and chitosan degree of acetylation (DA). These results revealed that the electrostatic interactions were the major driving force for chitosan–casein interaction, while hydrogen-bonding and hydrophobic interactions played a secondary role. It was also demonstrated that stable colloidal complexes only existed in the low casein concentration range [55]. More recently, we prepared chitosan–caseinate nanoparticles through polyelectrolyte complex (PEC) formation [56]. Such natural nanoparticles could be loaded with insulin for applications in controlled drug release. Chitosan and casein can also be associated with the layer-by-layer self-assembly technique [57] to create bioactive surfaces to enhance bio-mineralization, cell adhesion, and differentiation. Thus, chitosan–casein associations have been envisioned for various applications, including drug delivery systems and tissue engineering scaffolds, but not in detail for the processing of fibers for biomedical applications yet.

The production of multifunctional, bio-based, and biodegradable polymer fibers offers many advantages for biomedical applications. Our main objective is to establish the relationship between the processing, structure, and performance of chitosan–sodium caseinate-based materials. In this work, we propose to formulate and process, through wet spinning techniques, chitosan fibers containing sodium caseinate for the potential delivery of hydrophobic drugs. Our goal is to preserve the mechanical properties of the fibers for their further potential application in the development of knitted devices.

## 2. Results

The incorporation of sodium caseinate in chitosan fibers through wet spinning is described in the Materials and Methods section. In processing route R1, sodium caseinate was incorporated in the chitosan solution (the collodion). In processing route R2, sodium caseinate was incorporated in the coagulation bath with sodium hydroxide. The nomenclature of solutions, films, or fiber samples, thus, includes the processing route number and the concentration (*w*/*w*) (or weight fraction) of caseinate in the collodion *c*_1_ or in the coagulation bath *c*_2_.

### 2.1. Rheology of Chitosan/Sodium Caseinate Collodions

The flow diagrams obtained after pseudo-static (continuous) rheological tests on different mixed chitosan–caseinate collodions (thus related to processing route R1) are shown in Figure 1 for caseinate concentrations ranging from 0.04 to 7.2% (*w*/*w*). At a caseinate content of 7.2%, a gel-like behavior can be observed, as shown by a decrease in viscosity vs. shear rate with *η*~(dγ/dτ)^−1^, revealing plastic behavior. Such gels result from polyelectrolyte complexation of chitosan with caseinate [56] and partial neutralization of chitosan, yielding a gel at pH~6. The high viscosity did not allow easy extrusion and spinning from these compositions. At lower concentrations, however, the flow diagrams exhibited the usual behavior of polymer solutions with a Newtonian plateau in the low shear rate range and the flow regime associated with (chitosan) chain disentanglements. At lower concentrations, the influence of caseinate content on the viscosities of collodions is minimal, facilitating extrusion and fiber spinning within the pH range from 4.9 to 5.3 (see Appendix A for the evolution of the pH for different formulations prepared in route R1), caseinate tends to carry positive charges, fostering its dispersion in the chitosan solution and limiting the formation of polyelectrolyte complexes [37]. Thus, maintaining an acidic pH at moderate concentrations of caseinate was the way to keep a constant and reproducible rheological behavior of mixed solutions. In the following, we kept the same spinning parameters for the different processing routes R1 (in which the collodion contained a mixture of chitosan and caseinate) and R2 (chitosan coagulation bath was enriched with sodium caseinate). In the processing route R2, the collodion prepared for extrusion was always a pure chitosan solution at 5% (*w*/*w*), which was spun in a NaOH bath containing different mass fractions of caseinate to achieve chitosan–caseinate macro-filaments and dry fibers.

### 2.2. Evidence of Caseinate Incorporation Using ATR-FTIR Spectroscopy

The presence and semi-quantitative quantification of sodium caseinate within the different chitosan–caseinate formulations used for fiber spinning, with collodions obtained for the two processed routes R1 and R2, was performed with ATR-FTIR spectroscopy. In the first step, chitosan–caseinate film samples were processed at several known compositions, prepared from the neutralization of mixed chitosan/caseinate collodions, and their FTIR spectra were compared (Figure 2A). The absorption peaks of pure chitosan are present at 3370 cm^−1^ (O–H stretching overlapping the N–H stretching vibration). The presence of N-acetyl groups was confirmed by the bands at 1645 cm^−1^ (C=O stretching of amide I) and amine groups at 1551 cm^−1^ (–NH_2_), as shown by red arrows on the pure chitosan spectrum. Other characteristic bands appear at 1075 cm^−1^ (O–H stretching vibration), 1024 cm^−1^ (–C–O–C stretching vibration), and 930 cm^−1^. A C–O–C absorption band is present in the range from 1070 to 1080 cm^−1^ and amide II band at 1592 cm^−1^ (N–H stretching), whereas asymmetrical C–H bending of the CH_2_ group will manifest at 1485–1380 cm^−1^ [58]. The CH_2_ bending and CH_3_ symmetrical deformations were confirmed by the presence of bands at around 1420 cm^−1^ and 1375 cm^−1^, respectively, at 1026 cm^−1^ (C–O stretch, primary hydroxyl group) and 1060 cm^−1^ (C–O stretch, secondary hydroxyl group) as reported in the literature [59].

The sodium caseinate spectrum displays bands In the 1600–1500 cm^−1^, corresponding to the amide I and amide II bands, common to many proteins. Sodium caseinate powder displays an amide I band from 1660 cm^−1^, and films prepared from pure caseinate exhibited this band at 1640 cm^−1^. This shift for sodium caseinate films could indicate changes in the secondary structure after adsorption on the solid surfaces, as similar shifts were described for α-casein (10 cm^−1^) and for β-casein (16 cm^−1^) [59].

In the chitosan–caseinate mixtures, for the film samples prepared with different concentrations of sodium caseinate (*c*_1_ = 0.04%, 0.4%, 3.7%, and 7.2%), it can be seen that, as the caseinate content increases, the typical amide I band at 1640 cm^−1^ intensifies (see Figure 2A), confirming the presence of caseinate in the samples. Taking the absorption band at 1640 cm^−1^ for the pure caseinate sample as a reference, the relative composition of the formulations prepared at different caseinate mass fractions was established. Figure 3 shows that the absorbance at 1640 cm^−1^ increases with the increase in caseinate content in the chitosan–caseinate films prepared with the different caseinate mass fractions. Thus, the absorbance at 1640 cm^−1^ could be used as a caseinate probe to establish a calibration curve since the films were prepared with known compositions.

The same calibration curve was then used to estimate the caseinate content in the fibers obtained by the second processing route R2 (caseinate enrichment during coagulation step) through infrared analysis performed on a single fiber (see Figure 2B). The deduced compositions are in fair agreement with the nominal compositions of the coagulation baths, although this spectroscopy method only reveals the composition close to the external surface of fibers, in contact with the ATR crystal. Thus, the caseinate concentration, at least in the hydrogel periphery, tends to equilibrate with the coagulation baths. These results show that diffusion of sodium caseinate into the hydrogels was efficient for significant sodium caseinate loading into the fibers, in fact, comparable to the direct formulation of the caseinate-containing dope formulations (R1 route).

.

### 2.3. Fiber Microstructure Using SEM Analysis

The lateral surfaces of pure chitosan and CTS-R2 Cas 4% fibers were observed and compared using scanning electron microscopy (SEM) at different magnifications. SEM images (Figure 4) reveal that the surface morphology of the chitosan and CTS-R2 Cas 4% fibers exhibit a micron-range grooved/wrinkled surface, possibly resulting from the drying of the fibers (performed at constant length). At higher magnification, CTS-R2 Cas 4% fiber showed a homogenous distribution of white grains with a diameter close to 75 nm, whereas the pure chitosan fiber exhibited a smoother surface without these nanograins, ascribed to caseinate self-assemblies. Again, these observations show that caseinate was successfully integrated into the fibers (at least in their periphery). Such caseinate aggregates or micelles reformation was previously observed at high caseinate concentrations [60,61] and can be considered a positive issue for the performance of caseinate-loaded chitosan fibers, since micelles are known to favor the incorporation and delivery of various actives, including hydrophobic drugs [56,62,63].

Upon examination of fiber cross-sections, it was noticeable that the fracture surface of the chitosan fiber exhibits large smooth portions (see Figure 5), i.e., zones with a brittle rupture aspect. The rougher part of the surface shows a broken wooden ridge aspect that reveals the orientation of elementary chitosan fibrils in the yarn. However, in the case of CTS-R2 4% fibers, despite being prepared from a similar chitosan hydrogel, a rougher surface was observed. Again, this roughness may be attributed to the presence of caseinate aggregates, although fewer caseinate associations were found in the core of the fibers. This gradient of caseinate concentration could result from the diffusion process of caseinate within the hydrogel macro-filament, from the outer surface to the core, before their entrapment and aggregation during drying.

### 2.4. Crystalline Structural Analysis of Fibers using X-ray Diffraction

The influence of sodium caseinate on the crystalline arrangement of chitosan could be quantitatively evaluated by comparing the radial average curves of the two-dimensional wide-angle X-ray scattering patterns (2D-WAXS), as displayed in Figure 6. Figure 6A displays an example of a 2D-WAXS image of a chitosan–caseinate fiber (CTS-R2 Cas 10%) oriented vertically. The presence of the (020) and (200)+(220) reflexions in the equatorial sector (horizontally) evidences that the chain c axis of hydrated chitosan allomorph crystallites is oriented preferentially along the vertical fiber axis, as expected from the chitosan fiber wet spinning process including stretching to obtain good mechanical properties [23]. Figure 6B shows the radial average curves obtained for fibers of varied caseinate mass fractions after treatment and the corresponding 2D-WAXS images. Under the selected processing conditions, the incorporation of caseinate globally had a limited impact on the crystallinity of CTS-Cas fibers, while also maintaining the orientation of chitosan chains. The X-ray measurements display a slight increase in the amorphous phase with the introduction of caseinate in the dope to process the fibers through the mix collodion route, R1. For route R1, the Ph of the 5% chitosan solutions is close to the isoelectric point (IP) of caseinate and also below the apparent pKa of chitosan [37]. Both components thus carry positive charges limiting polyelectrolyte complexation through partial electrostatic repulsion in aqueous solutions. In the coagulation step, however, during the neutralization process, the increase in Ph due to NaOH diffusion in the chitosan–caseinate solution is relatively rapid so that the two associated components do not coexist for long periods as oppositely charged polymers. Thus, chitosan crystallization after neutralization is not perturbed by the establishment of polyelectrolyte complexes between chitosan and caseinate, mainly for kinetic reasons (and due to the high saline environment of the neutralization bath). Similarly, in the spinning route, R2, diffusion of NaOH may be fast in comparison with the diffusion of sodium caseinate, so that chitosan is neutralized before caseinate can establish polyelectrolyte complex interactions with chitosan, even if sodium caseinate is charged negatively at the Ph of the neutralization bath. As a result, in both processing routes, the establishment of caseinate–chitosan PECs is limited, and chitosan crystallization and drawing-induced orientation are preserved. We envision that, in processing route R1, PECs associations may be favored before spinning by increasing the Ph of mixed dopes above the IP of caseinate (4.6 to 5) but staying below the pKa of chitosan (~6.2) [64,65]. This Ph increase occurs when formulating the mixed collodions at high caseinate concentration (see Appendix A). In this Ph window, more electrostatic chitosan–caseinate interactions are expected, possibly resulting in a lower crystallization of chitosan and, thereby, a lower crystallinity of the fibers, which also could result in lower mechanical properties [23] if the viscosity of the resulting caseinate/chitosan dope is compatible with wet spinning.

### 2.5. Mechanical Properties of Chitosan–Caseinate Fibers

Typical stress–strain curves obtained through tensile testing of the different fibers are given in Figure 7, illustrating the behaviors of the systems prepared through the mixed collodion route, R1, and the coagulation enrichment route, R2, respectively.

All fibers exhibit a linear regime at low strains, yielding the elastic modulus and a plastic behavior at nominal strains above 1.5%. The introduction of large amounts of caseinate (≥0.4% (*w*/*w*)) in the collodions (route R1, see Figure 7A) induced a general decrease in mechanical properties, i.e., a decrease in the modulus, tenacity, and strain at break. These trends are reported quantitatively in Table 1, gathering all mechanical behavior data (N = 6 for each type of fiber).

However, for the samples prepared with coagulation enrichment (route R2), i.e., with caseinate incorporated in the coagulation bath (Figure 7A), a modest increase or preservation of mechanical properties, in comparison with pure chitosan, was obtained at caseinate concentrations up to 4% (*w*/*w*) in the coagulation bath (Figure 8).

As shown in the cross-property graphs in Figure 8, the fibers processed from route R2 keep better mechanical properties than the fibers formed with caseinate-containing collodions, located in the upper-right parts of the tenacity versus strain-at-break (σ_R_ vs. ε_R_) and Young’s modulus vs. strain-at-break (E vs. ε_R_) plots. Fibers processed from route R1 are located in the lower left zones, indicative of a significant decrease in mechanical properties with caseinate incorporation in the collodions. This could be comparatively due to better preservation of a denser chitosan network, pre-formed in the pure chitosan hydrogel macro-filament in processing route R2. Thus, the processing strategy consisting of a sodium caseinate enrichment of a pre-formed hydrogel would yield a significant loading of a pre-structured chitosan-based hydrogel, optimally preserving chitosan–chitosan interactions through the formation of crystallites acting as high functionality crosslinkers and entrapped entanglements in the amorphous phase.

Moreover, we used the tenacity values for the estimation of the Weibull’s modulus, deduced from the slopes of the Weibull diagrams displaying ${log}_{e}(-{log}_{e}(1-p))vs{log}_{e}(\sigma_{R})$, where the failure cumulative probability *p* is deduced from the ranks *i* of the values of $\sigma_{R}$ by the statistical correction *p* = (*i* − 0.3)/(*N* + 0.4). High values of the Weibull’s modulus are related with a narrow distribution of σ_R_ and, at a microscopic scale, to a homogeneity of critical defects, whereas the presence of a large distribution of defect sizes may induce a larger distribution of tenacity and lower values of the Weibull modulus [70]. In the fibers prepared from collodion mix (R1), the values of Weibull’s moduli were found to be significantly lower than in coagulation enrichment route R2 (see Table 1). This could be due to a better solubility of caseinate in alkaline conditions (hence the presence of lesser caseinate aggregates) and better-preserved chitosan–chitosan chain interactions when sodium caseinate is introduced in a pre-formed chitosan hydrogel in alkaline conditions.

## 3. Discussion

In the present study, the introduction of sodium caseinate in the chitosan collodions (mixed collodion route R1) or into the pre-formed chitosan hydrogel (coagulation enrichment route R2) generally induced a decrease in the mechanical properties of the fibers. However, for the samples prepared with caseinate incorporated in the coagulation step in alkaline conditions, ensuring caseinate submicelles dispersion, the resulting fibers displayed a preservation of their mechanical properties when we used concentrations of caseinate up to 4% (*w*/*w*). In route R1, where the caseinate was incorporated in the acidic chitosan dope, the loss of mechanical properties was clearer, even at a lower fraction of caseinate (0.4%), and could result from an alteration of the reformation of the hydrogel in with the formation of caseinate aggregates. Moreover, caseinate enrichment/loading of a pre-formed hydrogel is expected to affect less drastically the initial crystallization of chitosan to form the gel during neutralization. As a result, the semicrystalline microstructure of the dry fibers is less impacted by the presence of caseinate in the formulation, as shown by the X-ray diffraction structural analysis (see Figure 6B). Moreover, the introduction of caseinate had no significant impact on the orientation of chitosan crystals, which are known to be related to the tenacity of pure chitosan fibers [23,71].

The reduction in the ultimate mechanical properties of the fibers (particularly the deformation-at-break for fibers obtained from processing route R1) in the presence of larger amounts of caseinate could be related to the presence of caseinate aggregates (reformed micelles) acting as critical defects for fibers’ rupture. Thus, as a general trend, the impact of the protein incorporation process could be related to the initial solubility and homogeneity of the dispersion of the protein (depending on pH value), together with its ability to persist in the dry fiber in the form of single macromolecules or small sub-micellar objects.

Another possible mechanism for the entrapment of finely dispersed caseinate submicelles within the chitosan hydrogel in the coagulation enrichment route may also result from the ‘sieving effect’ of caseinate across the hydrogel. Large-size aggregates (in particular those forming at high caseinate concentration) may be more difficult to incorporate in the fine ‘porous’ or open structure of the hydrogel through a size exclusion effect, whereas individual submicelles may diffuse more efficiently. A more systematic microstructural study would be necessary to evidence more quantitatively the reformation of caseinate aggregates in mixed collodion route 1 at high values of caseinate content.

Consequently, in this work, the coagulation enrichment route is preferable, and yields loaded fibers with preserved mechanical properties, potentially acting as drug delivery systems when a drug is incorporated in the micelles.

## 4. Materials and Methods

### 4.1. Materials

Chitosan (CTS) was provided by Mahtani Chitosan Pvt. Ltd. (Veraval, India). The batch reference 114 (high molar mass and highly deacetylated chitosan from squid pen chitin) was fully characterized using size exclusion chromatography for the determination of weight-average molar mass (Mw) of 610 kg/mol and polydispersity index of D = 1.5, in agreement with the recommendations of the European Chitin Society (https://euchis.org/wp-content/uploads/2022/07/EUCHIS-Newsletter-No-49.pdf, access on 23 January 2024).

^1^H NMR spectroscopy was used to deduce the mean degree of acetylation DA (DA = 2%) according to the Hirai method [72]. Such low DA chitosan is known to exhibit good spinability and biocompatibility [23]. Sodium caseinate was kindly given by the Lactips company (Saint-Paul-en-Jarez, France). It is industrially produced from fat-free bovine milk. Casein was precipitated with hydrochloric acid at pH~4.6 and washed with 2M sodium hydroxide aqueous solution.

### 4.2. Preparations of Collodions and Coagulation Baths for Wet Spinning

To prepare the chitosan (CTS) solutions, chitosan powder was first swollen in distilled water for 1 h under mechanical stirring. Then, acetic acid was added in stoichiometric amounts to protonate the amine groups of chitosan and thereby achieve CTS dissolution in mild acidic conditions. The used chitosan concentration in the acid aqueous solution was fixed to 5% (*w*/*w*). In the first processing route, sodium caseinate powder was added to the mother chitosan solution, and water was added to obtain different caseinate concentrations *c*_1_ ranging from 0.04 to 7.2% (*w*/*w*), and a final chitosan concentration of *c*_chi_ = 3%. The obtained mixtures were kept under mechanical stirring for 12 h. Finally, collodions containing the mixture of chitosan and caseinate, or a ‘pure’ chitosan acetate solution used as reference, were obtained for fiber spinning. This fiber-spinning process will be hereafter referred to as the first processing route, R1.

In the second processing route, R2, caseinate solutions were prepared in 1M NaOH at different caseinate concentrations (or mass fractions) *c*_2_, in order to incorporate the caseinate into the coagulation bath used during the spinning of chitosan collodion. Thus, a pure CTS acetate solution (at a concentration of 5% *w*/*w*) was extruded for spinning in those conditions, where caseinate enrichment was achieved by diffusion of caseinate from the coagulation bath into the coagulated hydrogel filament.

The samples prepared through both processing routes, R1 and R2, and nomenclature are presented in Table 2.

### 4.3. Rheological Analysis on Collodions

An AR2000 rheometer (TA Instruments Ltd., New Castle, DE, USA) fitted with an aluminum Peltier plate, cone-plate geometry (diameter: 25 mm; angle: 4°) was used to characterize the rheological behavior of the different chitosan and chitosan–caseinate formulations at 25 °C. The experiments were performed with a gap size of 116 μm and a solvent trap to prevent partial drying of solutions. The analysis was performed in triplicate in continuous mode in a shear rate range from 10^−3^ to 10^2^ s^−1^. The flow diagrams, i.e., the plots of the steady-state shear viscosity vs. shear rate of the collodions, were obtained to deduce the Newtonian viscosity evaluated in the low shear rate range exhibiting a plateau, and the extrusion viscosities at a shear rate of about 10^2^ s^−1^.

### 4.4. Wet Spinning Process

We used a lab-scale wet spinning set-up as schematized in Figure 9. Extrusion was performed with a compressed air dispenser (Nordson Optimus, Nordson EFD, Feldkirchen, Germany). A controlled pressure, *P,* was applied onto a piston, inducing the extrusion of chitosan or chitosan–caseinate collodions through a 250, 410, or 580 μm conic spinneret (smooth flow tapered tips ref. 7018298, Nordson EFD, Feldkirchen, Germany,) at a linear velocity *V*_0_ = 20 mm·s^−1^. The extrudate was immersed in a coagulation bath (aqueous 1M NaOH, containing or not caseinate), allowing the neutralization of chitosan. Hydrogel macro-filaments were thus formed through alkaline gelation of chitosan. Three motorized bobbins and guides successively drove the coagulated macro-filaments in a washing bath (two baths with 5 L of deionized water) in a stretching and a drying zone. Drying was ensured in the airflow generated by a Mono 6 blower (Leister) associated with an LHS 21S heater (Leister) at a measured temperature of *T* = 140 °C. After the drying step, the monofilaments of the chitosan fibers were winded. Motorized reels allowed us to control the macro-filament’s linear speed at the exit of the neutralization bath, in the washing and in the drying zones (*V*_1_, *V*_2_, *V*_3_, respectively) and, thus, to impose the drawing of the filament in different locations. The impact of the repartition of the drawing ratio in the different spinning zones on mechanical properties was limited [23]. As a result, we only consider the impact of the total drawing ratio τ_tot_ = *V*_3_/*V*_0_ below. No drawing was applied in the coagulation bath; thus, τ_coag_ = *V*_1_/*V*_0_ = 1. The total drawing ratio τ_tot =_ τ_wash_ × τ_dry_ ranged from 1.62 to 1.67, with a constant value of τ_dry_ = *V*_3_/*V*_2_ = 1.3 (see Table 2).

### 4.5. X-ray Diffraction

Fibers were studied using X-ray diffraction at the European Synchrotron Facility (Grenoble France) on the BM2-D2AM beamline at an incident energy of 16 keV. The 2D scattered images were used to calculate the radial average around the image center using the Py-FAI library. The *q*-range calibration was performed using LaB6 powder.

### 4.6. Infrared Spectroscopy (ATR) on Films and Fibers

Films (with thicknesses of about 500 μm) were prepared by mimicking the fiber spinning process in processing route R1. This sample series, thus, resulted from the neutralization of chitosan–caseinate dope (at a chitosan concentration of 5% and different caseinate mass fractions) with NaOH 1M. These films, with a known amount of caseinate, were prepared using Teflon molds with circular recesses for the samples of 1 cm in diameter. An infrared spectrometer Nicolet iS10 (Thermofisher, Waltham, MA, USA) equipped with an ATR-diamond accessory was used. A total of 128 scans were recorded with a resolution of 4 cm^−1^ using an MCT/A detector in the wavenumber range from 4000 to 675 cm^−1^.

In order to study the fibers obtained in route R2, an infrared imaging microscope Nicolet iN10 (Thermofisher) was used in ATR configuration using a SlideOn MicroTip conical germanium crystal module with a tip diameter of 150 µm. A relative pressure of 30% of the built-in pressure monitoring sensor device was applied, while 256 scans were recorded with a resolution of 4 cm^−1^ using an MCT/A detector (cooled by liquid nitrogen) in the wavenumber range from 4000 to 675 cm^−1^. Three measurements were performed on a single fiber (at about 2.5 mm apart along the fiber axis). The obtained data, which were similar, were averaged.

In order to quantify the amount of caseinate present in the fibers, we established a calibration curve using the films obtained through the mixed collodion route R1, their known formulation composition, and the IR absorbance at 1640 cm^−1^. Then, the same calibration curve was used to evidence the presence of caseinate in the fibers obtained through the coagulation enrichment route R2, which allowed evaluation of the amount of caseinate within the fibers of this latter processing route.

### 4.7. Uniaxial Tensile Tests

Uniaxial tensile tests were performed using an AG-Xplus-10kN test machine (Shimadzu) equipped with a 100 N load cell. The fibers with an initial effective length of *L*_0_ = 50 mm were tested using an initial deformation rate of 1·min^−1^ (initial crosshead speed of 50 mm·min^−1^) as preconized by the ISO 5079:1995 standard [73]. The measurement of the force at break (*F*_r_) allowed calculating the uniaxial textile tenacity *T*e,_U_ = *F*_r_(N)/*Titre* (tex). The sample length at break (*L*_r_) allowed us to calculate the nominal strain-at-break ε_U,r_ = (*L*_r_ − *L*_0_)/*L*_0_. Mechanical characterizations were performed under ambient conditions, and the relative humidity was measured at *H*_r_ = (35 ± 5)%.

The *Titre* of monofilaments in tex (i.e., mass in grams of 1000 m of fiber) was calculated from mass measurements of a known length (>30 cm) of fiber. The theoretical mean diameter *D_th_* of the fibers was calculated from the *Titre* value according to Equation (1):(1)$$Titre=\frac{{\pi D}_{th}^{2}}{4}.\frac{\rho_{CTS}}{1000}\;$$
where *ρ_CTS_* = 1.4 g·cm^−3^ is the mass density of the chitosan fiber, and *D_th_* is the apparent diameter (in μm). The values deduced for *D_th_* were consistent with the direct estimations (*n* = 3) from the analyses of optical stereomicroscope images with LEICA M205 A equipment equipped with a LEICA DFC450 C camera. The image treatment was performed with LAS V4.12.

Microstructural analysis of fibers was performed with a FEI Quanta 250 scanning electron microscope (SEM) at an accelerating voltage of 5 kV. The beam current was reduced to avoid polymer degradation. Before SEM analyses, the sample was metalized by deposition of a thin layer of 10 nm thick copper. Chitosan fibers were observed under two different orientations, i.e., the lateral external surface or the fracture sections obtained by bending the fibers around a metallic needle immersed in liquid nitrogen.

## 5. Conclusions

The processing route for protein introduction in chitosan fibers has a significant impact on the mechanical properties, even though the semicrystalline microstructure of the fibers is not strongly decreased. The best mechanical properties of chitosan/protein fibers may be obtained when the presence of protein aggregates is avoided and when the initial semicrystalline structure of the chitosan hydrogel is preserved.

In order to show that the fibers obtained with chitosan and caseinate exhibit a higher loading capacity of hydrophobic drugs, further studies are necessary with micelles loaded with model drugs. Thus, a perspective is to optimize the composition ratios to find the proper tradeoff between the release of the drug and the preservation of mechanical properties.

Still, in the coagulation enrichment route, other strategies for improving the production of loaded chitosan fibers may consist of playing on the physicochemical context imposed on the chitosan hydrogel loaded with the protein of interest. After incorporation of caseinate in the hydrogel, decreasing the pH to 6–6.5 could still maintain the hydrogel and favor a negative charge on caseinate while chitosan would be partly protonated, inducing the formation, in situ within the gel, of electrostatic interactions (polyelectrolyte complexes) between both components.

In this work, the incorporation of caseinate through the coagulation enrichment of a pre-formed hydrogel led to achieving a more homogeneous distribution of caseinate in the fibers, leading to a reinforcement effect in the fiber. This approach should enable tuning or preserving the mechanical properties of the spun fibers and increase the mass ratio of caseinate, eventually by combining both caseinate incorporation routes to obtain efficient drug delivery by knittable fibers.

## Figures and Tables

**Figure 1 ijms-25-01768-f001:**
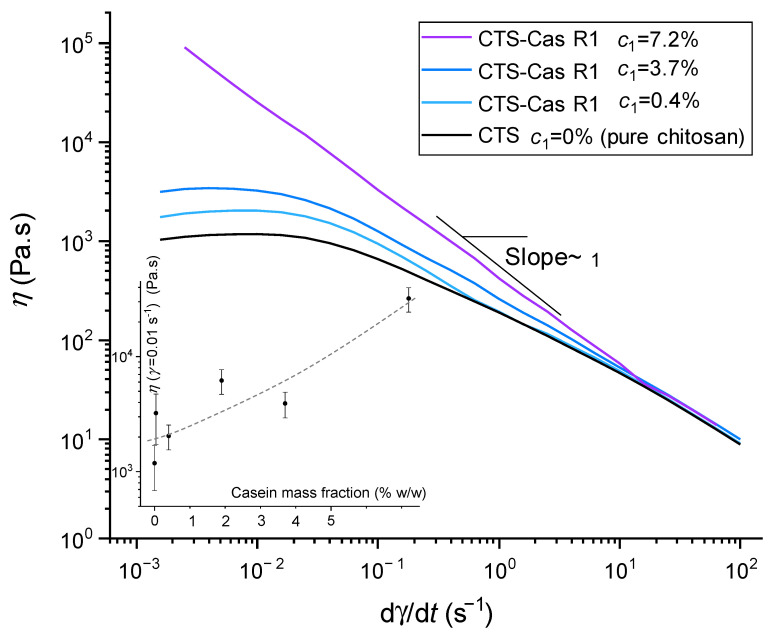
Examples of flow diagrams η vs. γ of chitosan/sodium caseinate collodions showing a Newtonian fluid behavior at a low shear rate and similar shear thinning behaviors above the shear rate $\dot{\gamma \;}$ = 10 s^−1^, with sodium caseinate concentration *c*_1_ ranging from 0 to 3.7% (*w*/*w*) and constant chitosan concentration of 3% (*w*/*w*). At a caseinate concentration of 7.2%, a pseudoplastic behavior is present with a decay slope ~1 in the log-log flow diagram, i.e., η~($\dot{\gamma }$)^−1^, evidencing the formation of a gel. The global increase in viscosity with caseinate content in mixed collodions is shown in the insert, displaying the evolution of the viscosity at $\dot{\gamma }\;$= 10^−2^ s^−1^. The dashed line is a guide for the eye, vertical error bars indicate +− standard deviation.

**Figure 2 ijms-25-01768-f002:**
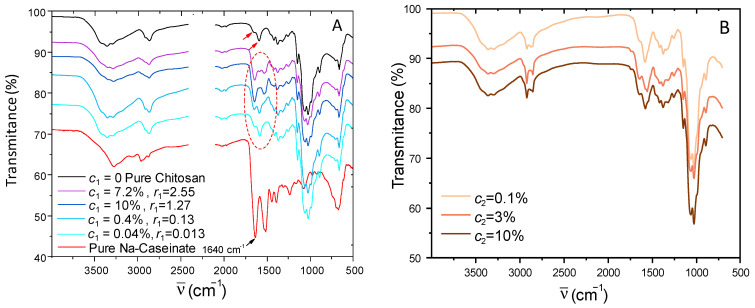
(**A**): Infrared spectra performed in ATR mode of pure chitosan, pure caseinate, and films obtained from mixed collodions, with sodium caseinate concentrations *c*_1_ = 0.04%, 0.4%, 3.7%, and 7.2% (*w*/*w*) and chitosan concentration 3% (*w*/*w*), neutralized in 1M NaOH aqueous solutions and then dried. The mass ratios *r*_1_ = mass of caseinate/mass of chitosan in the collodions and in final films are also shown. The red arrows display the bands of pure chitosan at 1645 cm^−1^ (C=O stretching of amide I) and amine groups at 1551 cm^−1^ (−NH_2_) (**B**): ATR infrared spectra from single fibers obtained through the coagulation enrichment route R2 (neutralization bath: 1M NaOH solution with sodium caseinate concentration *c*_2_ of 0.1%, 3%, and 10% (*w*/*w*)).

**Figure 3 ijms-25-01768-f003:**
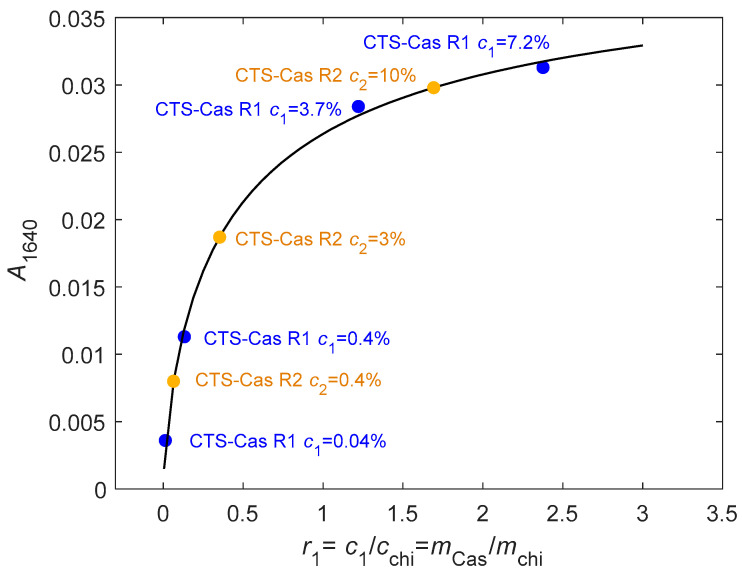
FTIR-ATR analyses used to estimate the sodium caseinate (Cas) content in the chitosan–caseinate fibers. (blue filled circle): Films obtained from mixed collodions with CTS−R1 (0.04%, 0.4%, 3.7%, and 7.2% *w*/*w* caseinate). (orange filled circle): Fibers obtained from coagulation enrichment route R2, namely CTS−R2 Cas 0.4%, 3%, and 10% were determined to contain ~0.15%, 0.4%, and 1.8% *w*/*w* caseinate at least at their surface. The absorption at 1640 cm^−1^ mainly reflects the sodium caseinate content, as the degree of acetylation of chitosan is very low (DA~2%, thus, with a low fraction of N-acetyl glucosamine residues). Black line modeling: $A_{1640}=a\frac{{r1}^{n}}{{r1}^{n}+b^{n}},\;a=0.0414,\;n=0.726,\;b=0.4595$.

**Figure 4 ijms-25-01768-f004:**
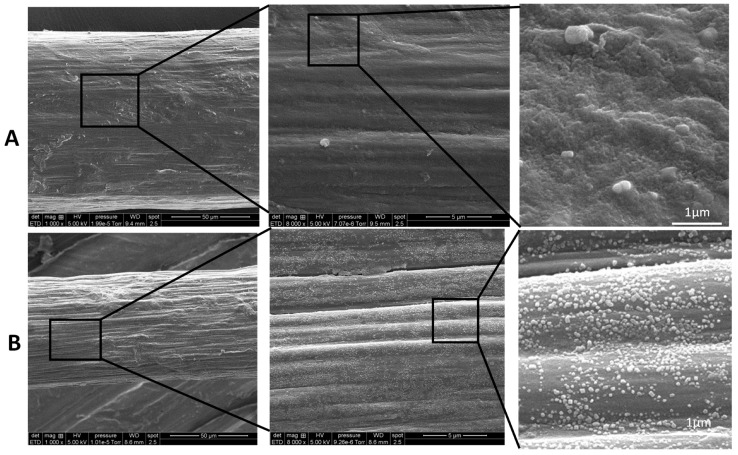
SEM micrographs of (**A**) Pure chitosan and (**B**) CTS−R2 Cas 4% fibers showing lateral surface.

**Figure 5 ijms-25-01768-f005:**
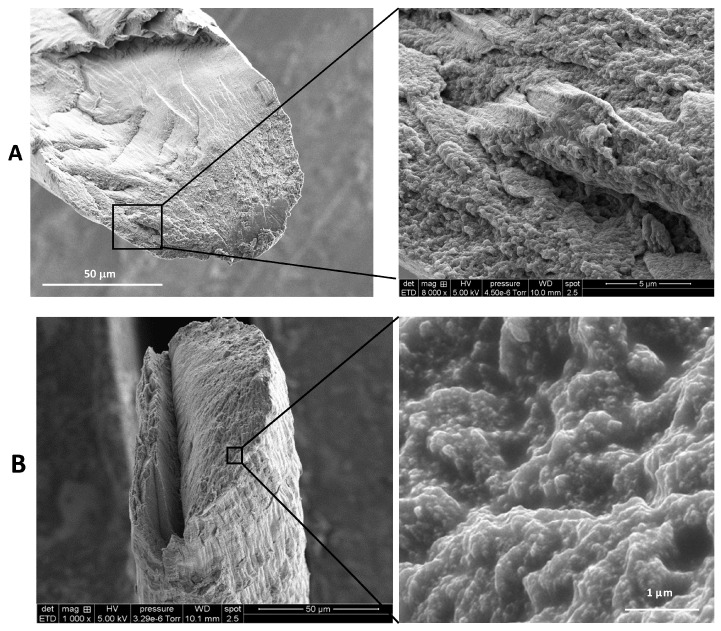
SEM micrographs at fracture surface of (**A**) pure chitosan fiber and (**B**) CTS−R2 Cas 4% fibers obtained from sodium hydroxide coagulation bath formulated with 4% *w*/*w* caseinate.

**Figure 6 ijms-25-01768-f006:**
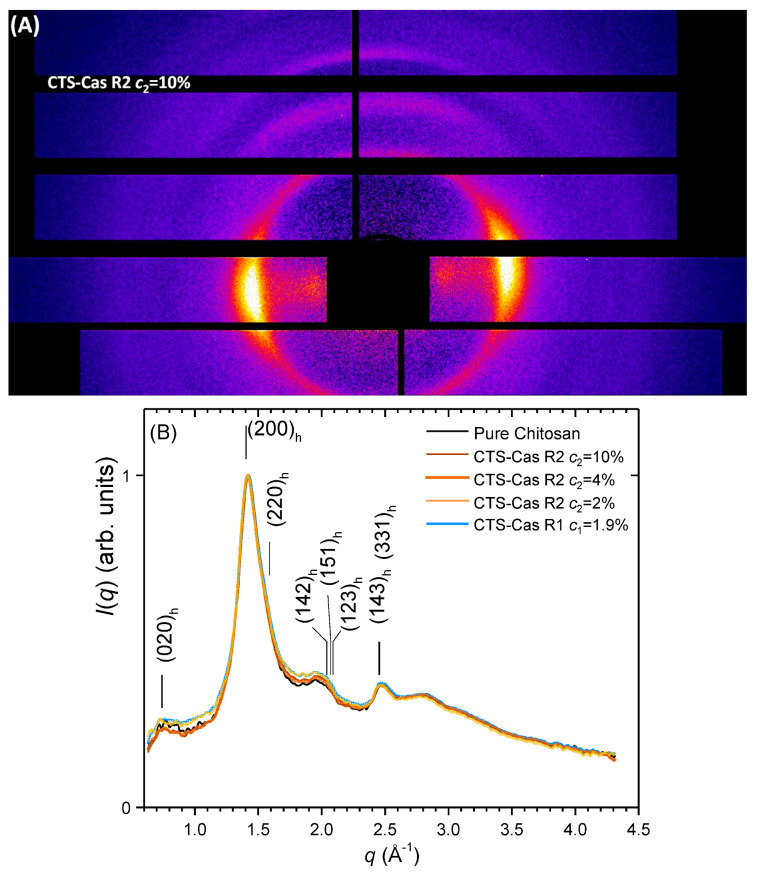
Microstructural X-ray analyses of pure chitosan and chitosan–caseinate fibers. (**A**) Example of two-dimensional wide-angle X-ray scattering image (2D-WAXS) of a chitosan–caseinate fiber (CTS R2 Cas 10%; route R2, *r*_2_ = 10%) showing anisotropy (obtained with the fiber axis being vertical). (**B**) Comparison of radial averages around the incident beam center for several fibers after normalization by the maximum intensity of each diffraction diagram. All fibers exhibit the hydrated allomorph of chitosan and no crystalline contribution of caseinate (indexation of chitosan allomorph reflections according to [66,67,68,69]). The caseinate introduction within the dope (route R1) or in the coagulation bath (route R2) only weakly impacts the crystalline ratio of chitosan. Pure chitosan does not contain sodium caseinate and, thus, displays a slightly higher crystalline ratio.

**Figure 7 ijms-25-01768-f007:**
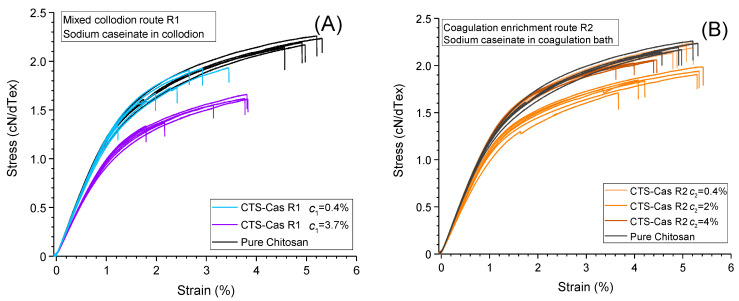
Examples of stress–strain plots for pure chitosan and mixed chitosan/caseinate fibers. All systems exhibit a linear elastic regime followed by plastic domain at applied strains above 1.5%. (**A**): Mixed collodion route R1, (**B**): Coagulation enrichment route R2.

**Figure 8 ijms-25-01768-f008:**
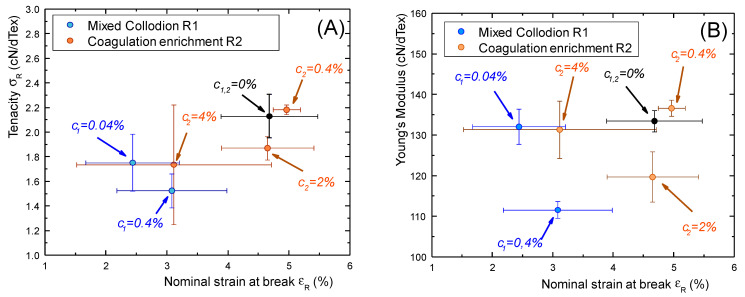
Crossed mechanical property plots of pure chitosan fibers and chitosan–caseinate fibers processed by following the different routes R1 (mix collodion route) and R2 (coagulation enrichment route). (**A**): Tenacity vs. deformation, showing better preservation of ultimate mechanical properties after R2 processing. (**B**): Young’s modulus vs. deformation at rupture showing again the location of R2 fibers in the upper-right part close to pure chitosan fiber properties.

**Figure 9 ijms-25-01768-f009:**
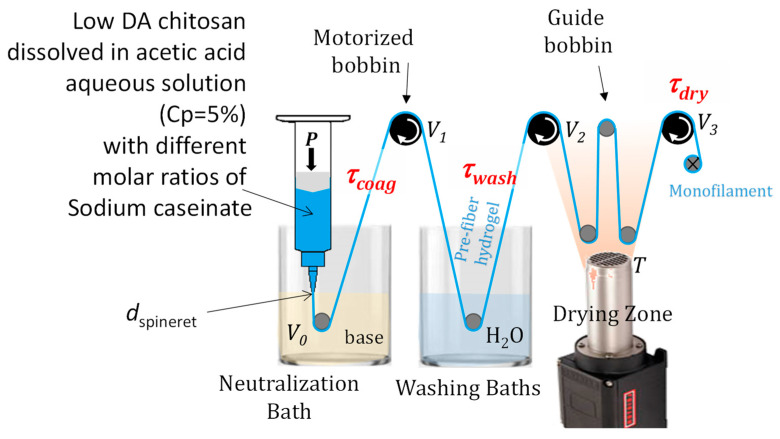
Scheme of the laboratory scale wet spinning process: the application of pressure *P* induces the extrusion of the dope dissolved at chitosan concentration *C*_p_ = 5% or 3% through a conic spinneret of final diameter *d*_spineret_ ranging from 250 to 580 μm, at an extrusion speed *V*_0_ in the coagulation bath (aqueous solution of NaOH 1M with various amounts of sodium caseinate), inducing the neutralization of chitosan. The macro-filament (‘prefiber’) is removed from the bath at a speed *V*_1_ (*V*_1_ = ω_1_·*R* where ω_1_ is the angular speed of the reel and *R* its radius) with a rotating bobbin and then washed in water baths by means of a second drive reel. Then, the hydrogel prefiber is driven into the drying zone by a third reel, where it is subjected to airflow at a temperature of *T*~140 °C before being winded as a dried monofilament. The path of the monofilament along the different zones is ensured by different guide grooves. The control of the extrusion speed and of the drive coils enables applying different drawing ratios in the coagulation bath (τ_coag_), in the washing zone (τ_wash_), and in the drying zone (τ_dry_). The product of the three drawing ratios is the total drawing ratio τ_tot_ = *V*_3_/*V*_0_.

**Table 1 ijms-25-01768-t001:** Mean mechanical properties (N = 6) of pure chitosan and chitosan–caseinate fibers obtained at 25 °C and 35% relative humidity (RH) and an initial deformation rate of 1 min^−1^.

Sample	Tenacity (cN/dtex) ± std	Young’s Modulus (cN/dtex) ± std	Strain-at-Break (%) ± std	Weibull’s Modulus m±5	Titer (Tex) ±2 Tex	Diameter (μm) from Titer and Density (1.4 g/cm^3^)
*c*_1_ = 0 (Pure Chitosan)	2.13 ± 0.18	133 ± 3	4.6 ± 0.79	35	50	70
	**Mixed collodion route R1 (Caseinate incorporated in the chitosan dope)**					
*c*_1_ = 0.1%	1.76 ± 0.23	132 ± 4	2.4 ± 0.77	7	58	81.2
*c*_1_ = 1%	1.52 ± 0.14	112 ± 2	3.1 ± 0.90	11	54	75.6
	**Coagulation enrichment route R2 (Caseinate incorporated in the neutralization bath)**					
*c*_2_ = 0.4%	2.18 ± 0.04	137 ± 2	5.0 ± 0.22	60	50	70.0
*c*_2_ = 2%	1.87 ± 0.1	120 ± 6	4.7 ± 0.76	20	63	88.2
*c*_2_ = 4%	1.74 ± 0.49	131 ± 7	3.1 ± 1.59	45	48	67.2

Colors indicate the nomenclature used in this work: blue = processing route R1, orange = processing route R2.

**Table 2 ijms-25-01768-t002:** Sample nomenclature, following two processing routes, R1 and R2. In the mixed collodion route R1, caseinate was introduced at a concentration *c*_1_ (*w*/*w*) in the chitosan acetate collodion. In the coagulation enrichment route R2, caseinate was incorporated in the coagulation bath at a mass fraction *c*_2_ to diffuse across the forming hydrogel macro-filament. These samples can be compared to a reference chitosan fiber (‘Pure CTS’) obtained with *c*_1_ = 0 or *c*_2_ = 0.

Mixed Collodion Route R1 (Caseinate Incorporated in the Chitosan Dope)			Coagulation Enrichment Route R2 (Caseinate Incorporated in the Coagulation Bath)		
Sample	*c*_1_ (*% w/w*) (Caseinate Mass Fraction in Dope)	Total Draw τ_tot_	Sample	*c*_2_ (*% w/w*) (Caseinate Mass Fraction in Coagulation Bath)	Total Draw τ_tot_
CTS-R1 Cas 0.04%	0.04%	1.67	CTS-R2 Cas 0.4%	0. 4%	1.67
CTS-R1 Cas 0.2%	0.2%	1.67	CTS-R2 Cas 2%	2%	1.67
CTS-R1 Cas 0.4%	0.4%	1.66	CTS-R2 Cas 3%	3%	
CTS-R1 Cas 1.9%	1.9%	1.62	CTS-R2 Cas 4%	4%	1.67
CTS-R1 Cas 3.7%	3.7%	1.67	CTS-R2 Cas 10%	10%	
CTS-R1 Cas 7.2%	7.2%				
Pure CTS	0	1.67	Pure CTS	0	1.67

Colors indicate the nomenclature used in this work: blue = processing route R1, orange = processing route R2.

## Data Availability

Data are contained within the article and Appendix A.

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
