# Peer review of "Wet-Spun Chitosan–Sodium Caseinate Fibers for Biomedicine: From Spinning Process to Physical Properties"

_ijms, 2024, doi:10.3390/ijms25031768_

Round 1
Reviewer 1 Report (Previous Reviewer 2)
Comments and Suggestions for Authors
Manuscript Title: Wet-Spun Chitosan-Casein Fibers for Biomedicine: from Spinning Process to Physical Properties
Manuscript ID: ijms-2791135.
It is clear that authors did a great effort revising their paper. They kindly addressed the issues that I requested. In addition, they added useful paragraphs and literature references, so the manuscript is now more consistent and well structured. Moreover, the spelling mistakes of the manuscript were checked and corrected, so as this version is well written in an easier language and more clear for readers. The plagiarism rate was also significantly decreased.
I suggest that the paper could be published in ijms journal in its present form.
Author Response
Reviewer 1 did not ask for any change.
Reviewer 2 Report (Previous Reviewer 1)
Comments and Suggestions for Authors
While the revised version submitted by the authors addressed most of the questions, some concerns remain with the overall manuscript.
For example, "amphiphilic" has been misunderstood and misinterpreted with hydrophobicity and hydrophilicity. Amphiphilic, by definition, has to be hydrophilic, oleophilic, and lyophilic with contact angles less than 90 degrees and a hysteresis less than 10 degrees. Casein is typically hydrophobic with a contact angle of more than 90 degrees. So, how is it amphiphilic?
How is Casein binding to chitosan fibers? Line 147 mentions "maintaining an acidic pH at moderate concentrations of casein..". What is the enzyme stability at acidic pH? There is no mention of the actual pH at which the experiments were carried out. So, we don't know precisely how Casein interacts because it doesn't seem to be a physical mixture, as the author claims in the following text.
Line 167: "Chitosan-casein film samples were processed at several known compositions from the mixed collodions," but most of their texts mention fibers. So, are these fibers or films? It isn't very clear from a materials science aspect.
It is almost impossible to detect the fingerprint zone at 930 cm-1 for the glycosidic bond) Use a Ge crystal and mid-IR instrument unless the instrument has near-IR capability. It's almost impossible to detect and decipher glycosidic bonds at low frequencies; they are considered unreliable data.
The central issue is how they detect and quantify separately the functional group fingerprints from Casein and Chitosan species since there is so much overlapping. The band at 1640 cm-1 (+/- 5 cm-1) from Casein comes from both the species, C=O stretching from amide (Chitosan) and pure Casein. This is where discussing the possible chemical intervention between these two species would have been very important.
Moreover, Casein is singular (e.g., Line 85 and in other places), not plural, as mentioned throughout most of the text.
While the mechanical evaluation and discussion are good, the composites' chemistry and materials science aspects deserve more attention to make their conclusions foolproof.
Comments on the Quality of English LanguageTypos and errors need to be taken care of.
Author Response
Reviewer 2
Q2.1. For example, "amphiphilic" has been misunderstood and misinterpreted with hydrophobicity and hydrophilicity. Amphiphilic, by definition, has to be hydrophilic, oleophilic, and lyophilic with contact angles less than 90 degrees and a hysteresis less than 10 degrees. Casein is typically hydrophobic with a contact angle of more than 90 degrees. So, how is it amphiphilic?
The vocabulary 'amphiphilic' was checked. In Merriam-Websters (https://www.merriam-webster.com/dictionary/amphiphilic)
--------------------------------------------------------------------------------------------------------------
Amphiphilic
adjective
am·-phi·-phil·-ic ËŒam(p)-fÉ™-ˈfi-lik
: of, relating to, or being a compound (such as a surfactant) consisting of molecules having a polar water-soluble group attached to a water-insoluble hydrocarbon chain
also : being a molecule of such a compound
"Chemists probably already know that these amphiphilic stabilizers are often a surfactant—a small molecule with a hydrophobic end and a hydrophilic end, such as those in detergents. — IEEE Spectrum, 16 Mar. 2022"
--------------------------------------------------------------------------------------------------------------
The word amphiphilic is indeed related to molecules with a hydrophilic and hydrophobic part. We used amphiphilic adjective in this sense, as commonly found in numerous works in literature to describe the properties of casein as “complex amphiphilic copolymers from mammalian milk”.
The search 'casein amphiphilic' on google yields 181 000 results, pointing out the specific role of beta-casein and K-casein for the structuration of casein complexes (micelles) due to their intrinsic amphiphilic nature.
As an example: "Casein micelles consist of cores of αs-caseinate covered by a uniform surface that contains αs and k-caseinate. The amphiphilic character is due to the presence of k-casein in the micelle surface, this protein consist in two-third hydrophobic part (N-terminal) and a third hydrophilic part". (R. J. Hill1969)
Addressing specifically the reviewer's definition of an amphiphilic molecule, casein was studied in previous works for the determination of contact angles and the surface energies of highly hydrated casein micelle layers (Britten 1989, Kühnl 2010) or casein films made from compressed particles (Guo 2021). All these published works show that the contact angles for water are lower than 90°C so that the amphiphile definition is actually met.
Q2.2. How is Casein binding to chitosan fibers?
In the conditions described in this work, we show that chitosan chain/ caseinate complexes may interact mildly, avoiding strong polyelectrolyte complexes. In the collodions used in route R1, the pH ensures that cationic repulsions should not drastically favor polyelectrolyte complexes between chitosan and casein micelles, and therefore precipitation. In the coagulation enrichment route, chitosan is mainly in the form of a neutral hydrogel, thus again with mild interactions with caseinate sub-micellar nanoparticles.
Q2.3. Line 147 mentions "maintaining an acidic pH at moderate concentrations of casein..". What is the enzyme stability at acidic pH? There is no mention of the actual pH at which the experiments were carried out. So, we don't know precisely how Casein interacts because it doesn't seem to be a physical mixture, as the author claims in the following text.
Many corrections were made in the text to answer this central comment.
First, casein and sodium caseinate are now strictly differentiated, in order to avoid confusion. This works deals with sodium caseinate, the stability of sodium caseinate was studied in many research works, and it was shown that sodium caseinate can re-aggregate in micelles at low pH and high concentration, or in presence of ions [Refs]. Since a sub-micellar organization is kept after casein treatment in sodium hydroxide, generating sodium caseinate, the system is never a ‘physical mixture’, but a suspension. Such argument led us not to use the word ‘dissolution’ to describe the mixing of chitosan solution and sodium caseinate (the word suspension, d
We measured the pH of all collodions and presented the results in table S1.
In the fibers prepared in this work, caseinate is able to reform aggregates in the acidic collodions or after drying of the fibers, as evidenced by electron microscopy. This shows that the coagulation conditions in Sodium hydroxide aqueous solutions permitted the reformation of caseinate pseudo-micelles, likely to be used for the loading of hydrophobic drugs. They should not be true casein micelles, since the original casein micelle organization is in fact lost by the conversion into sodium caseinate.
The physico-chemical behavior of casein and sodium caseinate is now better discussed, We also described the preparation of sodium caseinate from casein, as an illustration and for clarity.
Q2.4: Line 167: "Chitosan-casein film samples were processed at several known compositions from the mixed collodions," but most of their texts mention fibers. So, are these fibers or films? It isn't very clear from a materials science aspect.
This part was rewritten to better describe the establishment of the calibration curve thanks to films and the use of the calibration curve for the study of route 2 fibers. In particular the reference to a processing route (valid for fibers) is not used for films, so as to avoid confusion.
As an example:
" In preliminary studies, chitosan-casein film samples were processed at several known compositions from the mixed collodions of the route R1 and their FTIR spectra were compared (Figure 2A)."
For clarity, we changed this sentence to:
"In a first step, chitosan-casein film samples were processed at several known compositions, prepared from the neutralization of mixed chitosan/casein collodions and their FTIR spectra were compared (Figure 2A)"
In the same way, in the caption for figure 2, we corrected and enriched as follows:
" Figure 2. (A): Infrared spectra performed in ATR mode of pure chitosan, pure caseinate and films obtained from mixed collodions, with sodium caseinate concentrations c1= 0.04%, 0.4%, 3.7% and 7.2% (w/w), and chitosan concentration 3% (w/w), neutralized in 1M NaOH aqueous solutions and then dried. The mass ratios r1=mass of caseinate/mass of chitosan in the collodions and in final films are also shown. (B): ATR Infrared spectra from single fibers obtained through the coagulation enrichment route R2 (neutralization bath: 1M NaOH solution with a sodium caseinate concentration c2 of 0.1%, 3% and 10% (w/w)).
Q2.5 It is almost impossible to detect the fingerprint zone at 930 cm-1 for the glycosidic bond) Use a Ge crystal and mid-IR instrument unless the instrument has near-IR capability. It's almost impossible to detect and decipher glycosidic bonds at low frequencies; they are considered unreliable data.
Accordingly, the reference to the beta(1->4) glycosidic bond was removed.
Q2.6. The central issue is how they detect and quantify separately the functional group fingerprints from Casein and Chitosan species since there is so much overlapping. The band at 1640 cm-1 (+/- 5 cm-1) from Casein comes from both the species, C=O stretching from amide (Chitosan) and pure Casein. This is where discussing the possible chemical intervention between these two species would have been very important.
Indeed, overlapping is strong between casein and chitosan. The band at 1640 cm-1 is shown to be mainly related to caseinate content, since the degree of acetylation of chitosan is very low. Our chitosan is nearly poly glucosamine, thus there is no amide groups and no C=O stretching signature in this particular chitosan. This makes possible the calibration strategy with films. Accordingly, the figure cation of figure 3 was corrected and
This point was integrated in the figure caption of figure 3
Q2.7. Moreover, Casein is singular (e.g., Line 85 and in other places), not plural, as mentioned throughout most of the text.
This was corrected, except in one case when the different proteins constituting the casein particles are described. Casein was in fact replaced by sodium caseinate to avoid confusion.
Q2.8. While the mechanical evaluation and discussion are good, the composites' chemistry and materials science aspects deserve more attention to make their conclusions foolproof.
We thank the reviewer for his kind remark concerning the mechanical property evaluation and his constructive remarks to make the 'chemistry and material science aspects' more understandable.
We worked on the ‘Material parts’ includindg a forlumation table better explaining the fabrication processes, and as a general correction, we better discussed the results using sodium caseinate physico-chemistry.
Q2.9. Comments on the Quality of English Language: Typos and errors need to be taken care of.
We corrected many typos, but as usual, we rely on the editing service to detect remaining errors.
We decided to change the notations for the caseinate concentration as c1 and c2 (instead of r1 and r2) in order to avoid confusion with the names of the processing routes R1 and R2. We also have homogenized the nomenclature of samples and corrected all figures.
References
Britten, M., Boulet, M., Paquin, P. (1989). Estimation of casein micelles’ surface energy by means of contact angle measurements. Journal of Dairy Research, 56(02), 223. doi:10.1017/s0022029900026443
Guo Y., Wu C., Du M., Lin S, Xu X., Yu P., (2021),In-situ dispersion of casein to form nanoparticles for Pickering high internal phase emulsions, LWT, 139,doi: 10.1016/j.lwt.2020.110538.
Kühnl W., Piry A., Kaufmann V., Grein T., Ripperger S., Kulozik U., (2010) Impact of colloidal interactions on the flux in cross-flow microfiltration of milk at different pH values: A surface energy approach, Journal of Membrane Science, 352(1–2), 107-115, doi: 10.1016/j.memsci.2010.02.006
Hill R. J., Wake R. G., (1969) Amphiphile Nature of K-Casein as the Basis for its Micelle Stabilizing Property, Nature, 221
Schulte J., Stöckermann M., Gebhardt R., (2020) Influence of pH on the stability and structure of single casein microparticles, Food Hydrocolloids, 105, 105741, doi: 10.1016/j.foodhyd.2020.105741

Reviewer 3 Report (New Reviewer)
Comments and Suggestions for Authors
Dear Authors,
in your interesting manuscript, the following points should be added/changed to further improve it:
- If possible in this journal, please put Materials & Methods at the normal position, i.e. before the Results. Reading results without knowing what was done doesn't make sense.
- Fig. 1: What does the "1" inside the graph mean?
- Fig. 2a: Why is there a break in all lines, although there is no break in the x-axis? Please use identical font sizes in (a) and (b).
- Fig. 6b is missing the letter "b". Please use "arb. units" on the y-axis, this abbreviation is much more common than "un.".
- Fig. 7: Please use identical font sizes for the insets in (a) and (b) and a decimal point (instead of a comma) in "CTS-R1 0.1%").
- Table 1: Please use the normal "+-" sign instead of "+/-". Standard deviations cannot have more than 2 digits, averages have the same accuracies as the respective standard deviations.
- Fig. 9: Please don't change the font within one figure.
Author Response
Reviewer 3
Q3.1 in your interesting manuscript, the following points should be added/changed to further improve it:
- If possible in this journal, please put Materials & Methods at the normal position, i.e. before the Results. Reading results without knowing what was done doesn't make sense.
Indeed, the Materials and Methods could be place before the results, providing a much clearer presentation, but the structure of the template in IJMS required to switch this part later in the text.
We wrote a paragraph briefly presenting the material s and spinning processes in L135-145, referring to the Materials section.
Q3.3 - Fig. 1: What does the "1" inside the graph mean?
We changed the graph showing the 1 is the slope in log log scales, and introduced the explanation in the figure 1 caption.
Q3.4 - Fig. 2a: Why is there a break in all lines, although there is no break in the x-axis? Please use identical font sizes in (a) and (b).
The break is classical to hide useful peak due to CO2
Q3.5 - Fig. 6b is missing the letter "b". Please use "arb. units" on the y-axis, this abbreviation is much more common than "un.".
These improvements were introduced in the new figure versions.
Q3.5 - Fig. 7: Please use identical font sizes for the insets in (a) and (b) and a decimal point (instead of a comma) in "CTS-R1 0.1%").
The figures were reprocessed and the same font size is now used.
Q3.5 - Table 1: Please use the normal "+-" sign instead of "+/-". Standard deviations cannot have more than 2 digits, averages have the same accuracies as the respective standard deviations.
This was corrected in table 1.
Q3.5 - Fig. 9: Please don't change the font within one figure.
Some different fonts were needed as legend to explain nthe different elements of the setup.
Reviewer 4 Report (New Reviewer)
Comments and Suggestions for Authors
New chitosan-casein fibers processed by wet spinning were synthesized and characterized for biomedical applications. The idea is to find the best compromise between preserving the mechanical properties of the fibers and their appropriate composition for obtaining biomaterials with potential application in drug release. There are two ways of introducing casein into the chitosan structure: 1) directly into the polysaccharide dope or 2) spreads into the chitosan hydrogel from a coagulation bath containing sodium caseinate and sodium hydroxide. The latter route generated fibers with better mechanical properties due to the formation of a preformed polysaccharide gel enriched in casein which conserved the crystallinity ratio of chitosan structure. When casein was incorporated in the chitosan dope, the resulted fibers showed a lower modulus and lower mechanical properties compared to pure polysaccharide fibers, maybe due to a lower entanglement compactness in the amorphous phase of chitosan network.
The authors answered in detail at all the reviewers’ questions, completing the manuscript with the new details both in the body of the text and in the article graphs which led to the improvement of the article quality, thus making it publishable in IJMS. The topic is original. The conclusions are consistent with the arguments presented and they address the main question posed. The references are appropriate and the literature is up-to-date.I agree with the publication of the manuscript, after correcting the word drug instead of dug (line 556, page 17).
Author Response
Reviewer 4
Q4.1.The authors answered in detail at all the reviewers’ questions, completing the manuscript with the new details both in the body of the text and in the article graphs which led to the improvement of the article quality, thus making it publishable in IJMS. The topic is original. The conclusions are consistent with the arguments presented and they address the main question posed. The references are appropriate and the literature is up-to-date.
I agree with the publication of the manuscript, after correcting the word drug instead of dug (line 556, page 17).
The mistake was corrected
Round 2
Reviewer 2 Report (Previous Reviewer 1)
Comments and Suggestions for Authors
The authors have revised the manuscript with the suggested revisions. It can be accepted for publication in IJMS.
Comments on the Quality of English LanguageMinor typos, otherwise, fine.
This manuscript is a resubmission of an earlier submission. The following is a list of the peer review reports and author responses from that submission.
Round 1
Reviewer 1 Report
Comments and Suggestions for Authors
The authors present a decent property study on chitosan-casein wet spun fibers and implications of the fabrication method. While the work is interesting, I think some points should be addressed -
- In Fig 2 - Why has the plot for 0% casein not been included as a control? It is important to study how the incorporation of casein affects the collodion solutions' rheology compared to pure chitosan. Why does r1=1% have a lower viscosity? And why has r1=20% been excluded from the plot?
- It is unclear why the authors carried out different modes of ATR-FTIR for R1 and R2—keeping other parameters constant would allow for a better comparison.
- Why are SEM images of R1 fibers absent?
- Why have SEM images of only R2 been provided? And why only Cas 4%? The other concentrations should have been analyzed as well.
- Page 10 - 'Such presence of casein aggregates or micelles can be 347 considered as a positive issue for the performance of casein-loaded chitosan fibers, since 348 the micelles are known to favor the incorporation and delivery of various actives, 349 including hydrophobic drugs' - can a reference be provided for this statement?
- Can Figure 7a be explained in more detail? It is unclear what the takeaway from this image is.
- Why is the modulus of r2-4% comparatively lower? The authors justify this by the statement -The reduction of the ultimate mechanical properties of the fibers (particularly the 478 deformation at break) in the presence of larger amounts of casein could be related to the 479 presence of casein aggregates, acting as critical defects for fibers’ rupture.
SEM images of all the other fibers should be analyzed to verify this hypothesis.
- The authors claim that the mechanical properties of R2 are better, but the numbers are not all that different (Table 3). Can some more justification be provided?
- The sample preparation table (Table 1)shows ratios for R1 up to 20% and up to 10% for R2. But many of these samples have been excluded from the analysis in the results and discussions for rheology, XRD, and UTM (tensile) tests. Why?
- Elaboration on the potential applications of these materials should be provided in the conclusion.
Minor typos need to be taken care of
Reviewer 2 Report
Comments and Suggestions for Authors
Manuscript Title: Wet-Spun Chitosan-Casein Fibers for Biomedicine: from
Spinning Process to Physical Properties.
Manuscript ID: ijms-2561732.
The present study investigated the wet spinning of chitosan polysaccharide in combination with casein protein to produce biomedical fibers. The study focuses particularly on mechanical properties of designed chitosan/casein fibers for their potential applicability in the development of knitted textile biomedical devices.
The topic discussed in this paper is very interesting presenting obvious novelty and the experimental work is important. The language of the manuscript is good. Results found in this research work are of great interest.
Nevertheless, authors should address the below given points:
* Adding some explanations for experimental choices.
* Adding some curves to already existing figures.
* Decrease the plagiarism rate (actually 24 %). Particularly decrease auto-plagiarism (6%) from the reference : (Renaud Passieux et al. "Cytocompatibility / Antibacterial ActivityTrade-off for Knittable Wet-Spun ChitosanMonofilaments Functionalized by the In Situ Incorporation of Cu and Zn ", ACS Biomaterials Science & Engineering, 2022).
For detailed comments see the attached file: Review-ijms-2561732.pdf
